# Activation of General Control Nonderepressible-2 Kinase Ameliorates Glucotoxicity in Human Peritoneal Mesothelial Cells, Preserves Their Integrity, and Prevents Mesothelial to Mesenchymal Transition

**DOI:** 10.3390/biom9120832

**Published:** 2019-12-05

**Authors:** Theodoros Eleftheriadis, Georgios Pissas, Georgia Antoniadi, Evdokia Nikolaou, Spyridon Golfinopoulos, Vassilios Liakopoulos, Ioannis Stefanidis

**Affiliations:** Department of Nephrology, Faculty of Medicine, University of Thessaly, 41110 Larisa, Greece; g.antoniadi@yahoo.com (G.A.); nikolaoueyh@gmail.com (E.N.); spygolfin@yahoo.gr (S.G.); liakopul@otenet.gr (V.L.);

**Keywords:** peritoneal dialysis, mesothelial cell, GCN-2 kinase, glucose transporter, apoptosis, mesothelial to mesenchymal transition

## Abstract

Along with infections, ultrafiltration failure due to the toxicity of glucose-containing peritoneal dialysis (PD) solutions is the Achilles’ heel of PD method. Triggered by the protective effect of general control nonderepressible-2 (GCN-2) kinase activation against high-glucose conditions in other cell types, we evaluated whether the same occurs in human peritoneal mesothelial cells. We activated GCN-2 kinase with halofuginone or tryptophanol, and assessed the impact of this intervention on glucose transporter-1, glucose transporter-3, and sodium-glucose cotransporter-1, glucose influx, reactive oxygen species (ROS), and the events that result in glucotoxicity. These involve the inhibition of glyceraldehyde 3-phosphate dehydrogenase and the diversion of upstream glycolytic products to the aldose pathway (assessed by D-sorbitol), the lipid synthesis pathway (assessed by protein kinase C activity), the hexosamine pathway (determined by O-linked β-N-acetyl glucosamine-modified proteins), and the advanced glycation end products generation pathway (assessed by methylglyoxal). Then, we examined the production of the profibrotic transforming growth factor-β1 (TGF-β1), the pro-inflammatory interleukin-8 (IL-8). Cell apoptosis was assessed by cleaved caspase-3, and mesothelial to mesenchymal transition (MMT) was evaluated by α-smooth muscle actin protein. High-glucose conditions increased glucose transporters, glucose influx, ROS, all the high-glucose-induced harmful pathways, TGF-β1 and IL-8, cell apoptosis, and MMT. Halofuginone and tryptophanol inhibited all of the above high glucose-induced alterations, indicating that activation of GCN-2 kinase ameliorates glucotoxicity in human peritoneal mesothelial cells, preserves their integrity, and prevents MMT. Whether such a strategy could be applied in the clinic to avoid ultrafiltration failure in PD patients remains to be investigated.

## 1. Introduction

Peritoneal dialysis (PD) is one of the available options for treating patients with end-stage renal disease. Although it is as effective as hemodialysis [1], PD runs the risk of ultrafiltration failure, which could eventually lead to the patients switching to hemodialysis [2,3]. Ultrafiltration failure is accelerated by frequent episodes of peritonitis, yet is known to occur even in the absence of such potent inflammatory assaults [4]. 

Ultrafiltration failure may result from peritoneal membrane injury due to high glucose concentration, glucose degradation products (GDPs), and advanced glycation end-products (AGEs) in standard PD solutions [5,6,7,8]. The most prominent pathologic features of peritoneal membrane injury are the denudation of the mesothelial layer and submesothelial fibrosis. With years of PD treatment, the submesothelial space expands due to the accumulation of extracellular matrix [9]. Interestingly, the density of peritoneal capillary vessels, which correlates with ultrafiltration failure, increases with the broadening of the submesothelial space [9]. It should be noted that uremia per se contributes to alterations in the PD membrane. However, pathologic changes are accelerated with the application of PD. In a study, the median width of the submesothelial zone was 40 μm in normal individuals, 150 μm in uremic patients, and 150 μm in hemodialysis patients. In PD patients, the median thickness of the submesothelial collagenous zone progressed from 180 μm in those who have been on the method for less than two years to 600 μm in those who had undergone PD for more than eight years. The median thickness in PD patients with membrane failure was 675 μm [10].

The submesothelial extracellular matrix is produced by myofibroblasts, and most of them are derived by the mesothelial cells through the process of mesothelial to mesenchymal transition (MMT) [5,6,7]. Many studies confirmed that high glucose concentration, GDPs, and AGEs may induce MMT [5,6,7,8]. Inflammation also triggers MMT and explains the harmful impact of peritonitis episodes on the ultrafiltration capacity of the peritoneal membrane [4,5,6,7]. However, high-glucose PD solutions may also cause a low-grade inflammatory response in the peritoneal membrane [5,6,7]. Interestingly, myofibroblasts derived from mesothelial cells produce vascular endothelial growth factor-A (VEGF-A) [11,12], which may explain the correlation between submesothelial fibrosis and peritoneal capillary vessel density in patients with ultrafiltration failure [9]. 

A recent study in pediatric PD patients emphasizes the role of direct glucose toxicity over GDPs and AGEs, since it detected major early alterations in the peritoneal membrane of children treated with neutral pH and low-GDPs dialysis fluids [13]. Another interesting study has shown that the role of peritonitis may be overemphasized, and that PD vintage and glucose exposure drive PD membrane transformation [14]. 

Currently, attempts are underway to use glucose-free PD solutions [15]. Icodextrin is the most widely used glucose-free PD solution in the clinic; its use is associated with less deterioration of the peritoneal membrane function [16,17]. However, icodextrin is used in combination with glucose-containing PD solutions, and some data suggest that it may cause local inflammation [18]. Amino-acid-containing PD solutions are also in clinical use, but they should be applied at a ratio of 1 to 4 with glucose-containing PD solutions to avoid metabolic acidosis and increased serum nitrogen levels, and to achieve adequate ultrafiltration [19]. Also, their biocompatibility has not been thoroughly tested [20]. Other glucose-free PD solutions containing polyglycerol, xylitol, or taurine are under investigation at the preclinical level, with their safety, efficacy, and cost-effectiveness remaining to be clarified [20].

Thus, attempts to reduce glucotoxicity in mesothelial cells are essential since, if successful, they may prolong the effectiveness of PD. Fueled by previous data from our laboratory indicating that the activation of general control nonderepressible-2 (GCN-2) kinase decreases glucose influx in activated T-cells and capillary endothelial cells by downregulating the glucose transporter-1 (GLUT-1) [21,22], in the current study, we addressed the effect of this kinase on glucotoxicity, survival, and MMT in primary human peritoneal mesothelial cells cultured under high-glucose conditions. 

In endothelial cells, GCN-2 kinase activation reduces glucose influx in endothelial cells and prevents glucotoxicity since it decreases reactive oxygen species (ROS) overproduction, subsequent glyceraldehyde 3-phosphate dehydrogenase (GAPDH) inhibition, and the four known noxious pathways due to the accumulation of glycolytic products upstream of GAPDH [22]. Briefly, the four pathways encompass the polyol pathway that depletes the reduced form of nicotinamide adenine dinucleotide phosphate (NADPH) required for glutathione reduction increasing oxidative stress [23,24], the lipid synthesis pathway that produces diacylglycerol (DAG), which in its turn activates protein kinase C (PKC) [23,24], the conversion of accumulated glyceraldehyde 3-phosphate to methylglyoxal (MGO), a precursor of AGEs [23,24,25], and the hexosamine pathway that generates O-linked β-N-acetyl glucosamine (O-GlcNAc), which modifies many proteins, altering their function [26]. Interestingly, like capillary endothelial cells [22,27,28], mesothelial cells do not downregulate glucose transporters under high-glucose conditions; on the contrary, they increase their expression and glucose uptake [29].

We evaluated all the above pathways in mesothelial cells, starting from the effect of GCN-2 kinase activation on GLUT-1, GLUT3, and the sodium-glucose cotransporter 1 (SGLT1), since these glucose transporters have been identified in differentiated human mesothelial cells [30]. For the activation of GCN2 kinase, we used halofuginone. Halofuginone is already in use in veterinary medicine as a coccidiostat and has also been approved for the treatment of certain cases of scleroderma in humans. Halofuginone, by inhibiting propyl-tRNA synthetase, increases cellular uncharged tRNA molecules and activates GCN-2 kinase [31]. In its turn, the activated kinase phosphorylates the eukaryotic initiation factor-2α (e-IF2α), altering the protein translational program of the cell in the context of the cytoprotective integrated stress response [32]. Considering that, in addition to GCN-2 kinase activation, halofuginone interferes with transforming growth factor-β1 (TGF-β1) signal transduction [31] by inhibiting SMAD3 phosphorylation, we also evaluated the effect of tryptophanol, an experimental substance that activates GCN-2 kinase by inhibiting tryptophanyl-tRNA synthetase [22,33].

## 2. Materials and Methods

### 2.1. Cell Culture Conditions

Primary human peritoneal mesothelial cells (Zen-Bio Inc., Durham, NC, USA) were expanded in 75-cm^2^ flasks coated with gelatin (0.1% gelatin solution, ATCC primary cell solutions, LGC Standards GmbH, Wesel, Germany) in mesothelial cell growth medium (Zen-Bio Inc.). Passage 2 cells were used for the experiments.

Cells were cultured for 48 h in DMEM low glucose (5.5 mM D-glucose) (Thermo Fisher Scientific Inc., Rochford, IL, USA) or in DMEM high glucose (25 mM D-glucose) (Thermo Fisher Scientific Inc.), in the presence or not of 20 nM halofuginone (Cayman Chemicals, Ann Arbor, MI, USA) or 250 nM tryptophanol (Sigma-Aldrich; Merck Millipore, Darmstadt, Germany). It should be noted that DMEM low-glucose medium contains a normal glucose concentration. The above concentrations of halofuginone and tryptophanol were selected after assessing their cytotoxicity for mesothelial cells, as described below. Culture media were supplemented with 10% fetal bovine serum (Sigma-Aldrich; Merck Millipore) and antibiotics (penicillin 100 U/mL and streptomycin 100 μg/mL) (Sigma-Aldrich; Merck Millipore). In the culture medium of the cells cultured under normal glucose concentration (5.5 mM D-glucose), L-glucose (Sigma-Aldrich; Merck Millipore), which is not metabolized by the cells, was added to reach equal osmolarity with the high-glucose culture medium (25 mM D-glucose). 

For the experiments, mesothelial cells were cultured in six-well plates (30,000 cells per well), 24-well plates (50,000 cells per well), or 96-well plates (10,000 cells per well) in a humidified atmosphere containing 5% CO_2_ and at 37 °C. Each experiment was repeated nine times.

### 2.2. Assessment of Halofuginone and Tryptophanol Cytotoxicity

Mesothelial cells were cultured for 48 h in 96-well plates. Complete DMEM low glucose was used as the culture medium, and the cells were treated with or without escalating concentrations of halofuginone (10, 20 or 40 nM) or tryptophanol (125, 250, 500 nM). Cytotoxicity was assessed by the lactate dehydrogenase (LDH) release assay using the Cytotox Non-Radioactive Cytotoxic Assay kit (Promega Corporation, Madison, WI, USA), and calculated by the equation Cytotoxicity (%) = (LDH in the supernatant: Total LDH) × 100.

### 2.3. Assessment of the Proteins of Interest

Mesothelial cells were cultured in six-well plates under the described conditions. The T-PER tissue protein extraction reagent (Thermo Fisher Scientific Inc.), supplemented with protease and phosphatase inhibitors (Sigma-Aldrich; Merck Millipore), was used for cell protein extraction. The protein concentration of each cellular extract was assessed with a Bradford assay (Sigma-Aldrich; Merck Millipore), and 10 μg from each sample were used for Western blotting. Sodium dodecyl sulfate (SDS) polyacrylamide 4–12% Bis-Tris gels (Thermo Fisher Scientific Inc.) and polyvinylidene fluoride (PVDF) membranes (Thermo Fisher Scientific Inc.) were used for Western blotting.

Blots were incubated at 4 °C for 16 h with the primary antibodies, specific against the phosphorylated e-IF2α (p-eIF2α; 1:500; cat. no. 9721; Cell Signaling Technology, Danvers, MA, USA), GLUT-1 (1:200; cat. no. sc-7903, Santa Cruz Biotechnology, Dallas, TX, USA), GLUT-3 (1:100; cat no. sc-74497; Santa Cruz Biotechnology), SGLT-1 (1:100, cat. no. sc-98974; Santa Cruz Biotechnology), O-Glc-NAc-proteins (1:100; cat. no. sc59623; Santa Cruz Biotechnology), activated cleaved caspase-3 (1:1000, CC3; cat no. ab13847; Abcam, Cambridge, UK), α-smooth muscle actin (1:100; α-SMA; cat no. sc-130617; Santa Cruz Biotechnology), and β-actin (1:2500; cat. no. 4967; Cell Signaling Technology). Then, the secondary anti-rabbit IgG, HRP-linked antibody (1:1000; cat. no. 7074; Cell Signaling Technology) or anti-mouse IgG, HRP-linked antibody (1:1000; cat. no. 7076; Cell Signaling Technology) was applied for 30 min at room temperature. In the case of reprobing the PVDF blots, Restore Western Blot Stripping Buffer (Thermo Fisher Scientific Inc.) was used.

Bands were visualized with the LumiSensor Plus Chemiluminescent HRP substrate kit (GenScript Corporation, Piscataway, NJ, USA), and densitometric analysis of the bands was performed using ImageJ software (National Institute of Health, Bethesda, MD, USA).

### 2.4. Assessment of D-Glucose Consumption and ROS Production

Mesothelial cells were cultured in six-well plates. The Element Blood glucose monitor (Element, Infopia, Titusville, FL, USA) was used for measuring D-glucose in cell culture supernatants, and glucose consumption was calculated by subtracting the value of each measurement from the initial D-glucose concentration of the culture medium.

Mesothelial cells were cultured in 96-well plates for 48 h and, at the end of the period, 5 μM of the fluorogenic probe CellROX Deep Red Reagent (Invitrogen, Life Technologies, Carlsbad, CA, USA) were added to each well. Cells were incubated for 30 min at 37 °C and washed with phosphate-buffered saline (PBS) (Sigma-Aldrich; Merck Millipore) before measuring the fluorescence signal intensity with an EnSpire Multimode Plate Reader (Perkin Elmer, Waltham, MA, USA).

### 2.5. Assessment of GAPDH and PKC Activity

Before assessing GAPDH and PKC activities, a Bradford assay was performed, and the lysate volume of the samples was modified to equal the protein concentration.

For GAPDH activity, mesothelial cells were cultured in 24-well plates, and the activity of the enzyme was determined with a colorimetric GAPDH Assay (ScienCell, Carlsbad, CA, USA) according to the protocol provided by the manufacturer. 

For PKC activity, mesothelial cells were cultured in six-well plates, rinsed two times with ice-cold PBS and lysed with 1 mL lysis buffer. Following lysis, the PKC activity was measured colorimetrically with a PKC Kinase Activity Assay Kit (Abcam), according to the manufacturer’s instructions.

### 2.6. Assessment of D-Sorbitol and MGO Cellular Content

Mesothelial cells were cultured in six-well plates; once the experimental process was over, the cells were lysed with the T-PER tissue protein extraction reagent. A Bradford assay was performed in order to measure the protein concentration, and each lysate’s volume was adapted adjusted to contain 2 μg protein/mL. Cellular D-sorbitol and MGO content were assessed colorimetrically using the D-sorbitol Assay Kit (Abcam) and the Human Methylglyoxal ELISA Kit (MyBiosource, San Diego, CA, USA), respectively.

### 2.7. Assessment of TGF-β1 and IL-8 Production

TGF-β1 and IL-8 concentrations in the supernatants of six-well mesothelial cell cultures were measured colorimetrically using the Human TGF-beta-1 ELISA Kit (AssayPro, St. Charles, MO, USA), and the Human IL-8/NAP-1 ELISA Kit (Bender MedSystems GmbH, Vienna, Austria). The detection range of the above kits is 31–2000 pg/mL and 15.6–1000 pg/mL, respectively.

### 2.8. Statistical Analysis

IBM SPSS Statistics for Windows, version 20 (IBM Corp., Armonk, NY, USA), was used for the statistical analysis. One sample Kolmogorov–Smirnov test confirmed that the evaluated variables were normally distributed. For comparison of means, one-way analysis of variance (ANOVA) was used, followed by Bonferroni’s correction test. Results were expressed as mean ± standard error of means (SEM) and *p* < 0.05 was considered statistically significant. 

## 3. Results

### 3.1. Both Halofuginone and Tryptophanol, at Nontoxic Concentrations, Activate GCN2 Kinase

Mesothelial cells were cultured under normal glucose in the presence or not of escalated concentrations of tryptophanol (125, 250, 500 nM) or halofuginone (10, 20, 40 nM). Tryptophanol exerted toxicity only at the concentration of 500 nM (Figure 1A), whereas halofuginone was cytotoxic for mesothelial cells only at a concentration of 40 nM (Figure 1B). The maximum confirmed nontoxic concentration of the above substances was used for all the subsequent experiments, with tryptophanol used at a concentration of 250 nM, and halofuginone at 20 nM.

Next, mesothelial cells were cultured under normal or high-glucose conditions in the presence or not of 250 nM tryptophanol or 20 nM halofuginone. The capacity of the above substances at the used concentrations to activate GCN-2 kinase was evaluated by the level of phosphorylation of the GCN-2 kinase substrate e-IF2α. Nine such experiments were performed for each substance; three of them are depicted in Figure 1C,E.

High glucose left the p-eIF2α level unaffected. Tryptophanol enhanced the p-eIF2α level both under normal glucose (optical density (OD) 12.70 ± 0.88 vs. 4.83 ± 0.42, *p* < 0.05), and high glucose (OD 10.98 ± 0.62 vs. 4.81 ± 0.16, *p* < 0.05) (Figure 1D). Similarly, halofuginone increased the p-eIF2α level both under normal glucose (OD 12.07 ± 0.49 vs. 3.75 ± 0.35, *p* < 0.05), and high glucose (OD 13.75 ± 0.96 vs. 3.76 ± 0.37, *p* < 0.001) (Figure 1F).

### 3.2. In Mesothelial Cells Cultured under High-Glucose Conditions, Halofuginone Reduces the Degree of GLUT-1, GLUT-3 and SGLT-1 Increment, and Tryptophanol Exerts a Similar Effect with the Exception of GLUT-3

Mesothelial cells were cultured under normal or high-glucose conditions, in the presence or not of 250 nM tryptophanol or 20 nM halofuginone, and the expression of GLUT-1, GLUT-3, and SGLT-1 was assessed with Western blotting. Nine such experiments were performed for each substance; three of them are depicted in Figure 2A,E.

The GLUT-1 ODs obtained from normal glucose, normal glucose with halofuginone, high glucose, and high glucose with halofuginone were 8.20 ± 0.47, 4.12 ± 0.07, 14.93 ± 0.47, and 6.07 ± 0.66, respectively. Compared to the control, high glucose raised the GLUT-1 expression (*p* < 0.05), while halofuginone reduced GLUT-1 under both normal glucose (*p* < 0.05), and high-glucose conditions (*p* < 0.01) (Figure 2B). Similar were the results of the experiments with tryptophanol. The GLUT-1 ODs obtained from normal glucose, normal glucose with tryptophanol, high glucose, and high glucose with tryptophanol were 10.34 ± 0.83, 4.99 ± 1.21, 13.91 ± 0.69, and 4.10 ± 0.74, respectively. Compared to the control, high glucose enhanced GLUT-1 expression (*p* < 0.05), while tryptophanol reduced GLUT-1 under both normal glucose (*p* < 0.05) and high-glucose conditions (*p* < 0.01) (Figure 2F).

The GLUT-3 ODs obtained from normal glucose, normal glucose with halofuginone, high glucose, and high glucose with halofuginone were 8.67 ± 0.62, 5.22 ± 0.10, 13.35 ± 0.48, and 6.08 ± 0.32, respectively. Compared to the control, high glucose increased GLUT-3 expression (*p* < 0.05), while halofuginone reduced GLUT-3 under both normal (*p* < 0.05) and high-glucose conditions (*p* < 0.05) (Figure 2C). However, in contrast to halofuginone, the data from experiments with tryptophanol were different. The GLUT-3 ODs obtained from normal glucose, normal glucose with tryptophanol, high glucose, and high glucose with tryptophanol were 5.46 ± 0.30, 5.83 ± 0.32, 10.87 ± 0.25, and 11.18 ± 0.18, respectively. Compared to the control, high glucose upregulated GLUT-3 expression (*p* < 0.05), but tryptophanol did not alter GLUT-3 lever under either normal or high-glucose conditions (Figure 2G). 

The SGLT-1 ODs obtained from normal glucose, normal glucose with halofuginone, high glucose, and high glucose with halofuginone were 7.74 ± 0.48, 3.51 ± 0.16, 15.47 ± 0.58, and 6.60 ± 0.65, respectively. Compared to the control, high glucose increased SGLT-1 expression (*p* < 0.05), while halofuginone reduced SGLT-1 under both normal (*p* < 0.05), and high-glucose conditions (*p* < 0.05) (Figure 2D). The results from experiments with tryptophanol were somewhat different. The SGLT-1 ODs obtained from normal glucose, normal glucose with tryptophanol, high glucose, and high glucose with tryptophanol were 3.99 ± 0.46, 3.96 ± 0.46, 14.62 ± 0.56, and 10.76 ± 0.50, respectively. Compared to the control, high glucose upregulated SGLT-1 expression (*p* < 0.05). Tryptophanol did not change SGLT-1 under normal glucose, but decreased SGLT-1 under high-glucose conditions (*p* < 0.05), albeit to a lesser extent than halofuginone (Figure 2H). 

### 3.3. In Mesothelial Cells Cultured under High-Glucose Conditions, Both Tryptophanol and Halofuginone Decrease the Degree of Enhanced Glucose Consumption and ROS Production 

Glucose consumption under normal glucose, normal glucose with tryptophanol, normal glucose with halofuginone, high glucose, high glucose with tryptophanol, and high glucose with halofuginone was 76.33 ± 5.53, 74.33 ± 3.71, 59.00 ± 6.23, 211.00 ± 8.79, 133.36 ± 4.11, and 66.67 ± 2.60 mg/dL, respectively. Glucose consumption increased under high-glucose conditions (*p* < 0.05). Both tryptophanol (*p* < 0.05) and halofuginone (*p* < 0.05) significantly decreased the consumption of glucose by mesothelial cells cultured in a high-glucose medium, halofuginone to a greater extent than tryptophanol (*p* < 0.05) (Figure 3A).

ROS production, measured in fluorescence signal intensity units, under normal glucose, normal glucose with tryptophanol, normal glucose with halofuginone, high glucose, high glucose with tryptophanol, and high glucose with halofuginone was 22.00 ± 1.15, 20.67 ± 1.17, 20.66 ± 0.83, 56.33 ± 1.20, 31.68 ± 1.09, and 29.00 ± 1.61, respectively. ROS production increased under high-glucose conditions (*p* < 0.05). Tryptophanol reduced the production of ROS by mesothelial cells cultured in high glucose medium (*p* < 0.05), and halofuginone exerted a similar effect (*p* < 0.05) (Figure 3B).

### 3.4. In Mesothelial Cells Cultured under High-Glucose Conditions, Both Tryptophanol and Halofuginone Prevent GAPDH Inhibition and Ameliorate the High-Glucose-Induced Noxious Pathways

GAPDH activity under normal glucose, normal glucose with tryptophanol, normal glucose with halofuginone, high glucose, high glucose with tryptophanol, and high glucose with halofuginone was 97.33 ± 1.76, 98.67 ± 3.76, 94.35 ± 2.46, 50.66 ± 0.88, 105.30 ± 3.44, and 93.00 ± 2.31 mU/μg of protein, respectively. High glucose inhibited GAPDH activity (*p* < 0.05), while both tryptophanol (*p* < 0.05) and halofuginone (*p* < 0.05) restored GAPDH activity to the levels detected under normal glucose (Figure 4A).

Cellular D-sorbitol content under normal glucose, normal glucose with tryptophanol, normal glucose with halofuginone, high glucose, high glucose with tryptophanol, and high glucose with halofuginone was 6.55 ± 0.24, 7.20 ± 0.15, 7.44 ± 0.26, 22.07 ± 0.53, 11.86 ± 0.14, and 9.91 ± 0.05 nM, respectively. High glucose enhanced cellular D-sorbitol (*p* < 0.05), while both tryptophanol (*p* < 0.05), and halofuginone (*p* < 0.05) ameliorated this high-glucose-induced change (Figure 4B).

PKC activity under normal glucose, normal glucose with tryptophanol, normal glucose with halofuginone, high glucose, high glucose with tryptophanol, and high glucose with halofuginone was 0.63 ± 0.04, 0.63 ± 0.01, 0.75 ± 0.02, 2.30 ± 0.10, 1.48 ± 0.07, and 1.18 ± 0.04 OD/100 μg of protein, respectively. High glucose increased PKC activity (*p* < 0.05). Both tryptophanol (*p* < 0.05) and halofuginone (*p* < 0.05) decreased PKC activity in mesothelial cells cultured in high-glucose medium, halofuginone to a greater extent than tryptophanol (*p* < 0.05) (Figure 4C). 

Cellular MGO under normal glucose, normal glucose with tryptophanol, normal glucose with halofuginone, high glucose, high glucose with tryptophanol, and high glucose with halofuginone was 29.00 ± 0.76, 35.33 ± 0.93, 24.34 ± 0.73, 119.00 ± 7.00, 66.00 ± 1.62, and 37.30 ± 0.84 ng/mL, respectively. High glucose induces cellular MGO generation (*p* < 0.05). Both tryptophanol (*p* < 0.05) and halofuginone (*p* < 0.05) decreased MGO in mesothelial cells cultured in a high-glucose medium—halofuginone to a greater extent than tryptophanol since it restored MGO to the levels detected under normal glucose (*p* < 0.05) (Figure 4D).

The O-GlcNAc-proteins level was evaluated with Western blotting, and three of the nine experiments performed with halofuginone or tryptophanol are depicted in Figure 5A,C, respectively. The O-GlcNAc-proteins ODs obtained from normal glucose, normal glucose with halofuginone, high glucose, and high glucose with halofuginone were 6.25 ± 0.61, 6.16 ± 0.79, 13.05 ± 0.83, and 7.87 ± 0.98, respectively. Compared to the control, high glucose increased the level of O-GlcNAc-proteins (*p* < 0.05), while halofuginone reduced O-GlcNAc-proteins to the level detected in mesothelial cells cultured under normal glucose conditions (*p* < 0.05) (Figure 5B). Similar were the results of the experiments with tryptophanol. The O-GlcNAc-proteins ODs obtained from normal glucose, normal glucose with tryptophanol, high glucose, and high glucose with tryptophanol were 7.51 ± 0.67, 5.88 ± 0.53, 14.29 ± 1.00, and 5.64 ± 0.85, respectively. Compared to the control, high glucose promoted the generation of O-GlcNAc-proteins (*p* < 0.05), while tryptophanol restored O-GlcNAc-proteins to the level observed under normal glucose (*p* < 0.05) (Figure 5D).

### 3.5. In Mesothelial Cells Cultured under High-Glucose Conditions, Both Tryptophanol and Halofuginone Reduce the Degree of the TGF-β1 and IL-8 Increment

The concentration of TGF-β1 in cell culture supernatants under normal glucose, normal glucose with tryptophanol, normal glucose with halofuginone, high glucose, high glucose with tryptophanol, and high glucose with halofuginone was 48.35 ± 2.49, 34.00 ± 1.62, 38.67 ± 1.36, 395.50 ± 17.50, 225.00 ± 10.61, and 137,67 ± 3.19 pg/mL, respectively. High glucose upregulated TGF-β1 production (*p* < 0.05), while both tryptophanol (*p* < 0.05) and halofuginone (*p* < 0.05) reduced the degree of TGF-β1 increment, halofuginone to a greater extent than tryptophanol (*p* < 0.05) (Figure 6A).

The concentration of IL-8 in cell culture supernatants under normal glucose, normal glucose with tryptophanol, normal glucose with halofuginone, high glucose, high glucose with tryptophanol, and high glucose with halofuginone was 25.00 ± 1.04, 29.00 ± 1.04, 28.33 ± 1.64, 146.00 ± 6.64, 54.00 ± 5.41, and 97.00 ± 3.22 pg/mL, respectively. High glucose upregulated IL-8 production (*p* < 0.05), while both tryptophanol (*p* < 0.05) and halofuginone (*p* < 0.05) decrease the degree of IL-8 rise, tryptophanol to a greater extent than halofuginone (*p* < 0.05) (Figure 6B).

### 3.6. Both Tryptophanol and Halofuginone Prevent High-Glucose-Induced Cellular Apoptosis and the Mesothelial to Mesenchymal Transition

Cellular apoptosis was evaluated by the level of activated cleaved caspase-3, in which all the apoptotic pathways converge, and MMT by the level of the myofibroblast differentiation marker α-SMA. Three of the nine experiments performed with halofuginone or tryptophanol are depicted in Figure 7A,D, respectively.

The cleaved caspase-3 ODs obtained from normal glucose, normal glucose with halofuginone, high glucose, and high glucose with halofuginone were 5.19 ± 0.59, 5.39 ± 0.56, 15.55 ± 0.85, and 7.20 ± 0.39, respectively. Compared to the control, high glucose increased the level of cleaved caspase-3 (*p* < 0.05), while halofuginone reduced cleaved caspase-3 to the level detected in mesothelial cells cultured under normal glucose (*p* < 0.05) (Figure 7B). Similar were the results of the experiments with tryptophanol. The cleaved caspase-3 ODs obtained from normal glucose, normal glucose with tryptophanol, high glucose, and high glucose with tryptophanol were 6.95 ± 0.42, 5.39 ± 0.34 13.33 ± 0.67, and 7.66 ± 0.13, respectively. Compared to the control, high glucose triggered cell apoptosis (*p* < 0.05), while tryptophanol restored cleaved caspase-3 to the level observed under normal glucose (*p* < 0.05) (Figure 7E).

The α-SMA ODs obtained from normal glucose, normal glucose with halofuginone, high glucose, and high glucose with halofuginone were 6.62 ± 0.34, 6.93 ± 0.33, 13.15 ± 0.45, and 6.64 ± 0.05, respectively. Compared to the control, high glucose increased the level of α-SMA (*p* < 0.05), while halofuginone reduced α-SMA to the level detected in mesothelial cells cultured under normal glucose (*p* < 0.05) (Figure 7C). Regarding the experiments with tryptophanol, α-SMA ODs obtained from normal glucose, normal glucose with tryptophanol, high glucose, and high glucose with tryptophanol were 6.60 ± 0.63, 5.88 ± 0.52, 12.04 ± 0.86, and 8.81 ± 0.91, respectively. Compared to the control, high glucose promoted MMT (*p* < 0.05), while tryptophanol decreased the high-glucose-induced rise of α-SMA significantly, albeit not to the level detected under normal glucose (*p* < 0.05) (Figure 7F).

## 4. Discussion

Along with infections, ultrafiltration failure is the Achilles’ heel of PD, being the result of peritoneal membrane alterations due to exposure to the high glucose concentration, GDPs, and AGEs of the commonly used PD solutions [5,6,7]. Denudation of peritoneal membrane from mesothelial cells, as well as submesothelial fibrosis, are the prominent pathological findings [9]; interestingly, in patients with ultrafiltration failure, the latter is associated with the density of peritoneal membrane capillary vessels [9]. It is thought that submesothelial fibrosis is owed mainly to MMT [5,6,7].

Similar to capillary endothelial cells [22,27,28], mesothelial cells upregulate glucose transporters and uptake under high-glucose conditions [29]. Triggered by the protective effect of GCN-2 kinase activation against glucose influx and glucotoxicity in endothelial cells [22], we evaluated whether activation of this kinase ameliorates glucotoxicity in primary human peritoneal mesothelial cells, preserves their integrity, and prevents MMT.

First, we confirmed that both halofuginone and tryptophanol at nontoxic concentrations activate GCN-2 kinase, as evaluated by the level of the phosphorylation of its substrate, e-IF2α [32]. Next, we showed that high glucose increases the expression of the glucose transporters, known to be expressed in human peritoneal mesothelial cells, i.e., GLUT-1, GLUT-3, and SGLT-1 [30]. Halofuginone prevented the upregulation of the above glucose transporters, while tryptophanol prevented the upregulation of GLUT-1, reduced the upregulation of SGLT-1, and left the expression of GLUT-3 unaffected. 

The decrease in the expression of glucose transporters in mesothelial cells cultured under high-glucose conditions by halofuginone and tryptophanol reduced glucose uptake by the cells. This was more marked with halofuginone than with tryptophanol, possibly due to the more significant downregulation of the aforementioned glucose transporters by the first substance. 

Under high-glucose conditions, in the context of endothelial cells, the increased glucose influx due to the upregulation of GLUT-1 results in enhanced ROS production. The catabolism of large quantities of glucose through the Krebs cycle results in a vast supply of electron donors (NADH and FADH_2_) that overwhelms the capacity of oxidative phosphorylation chain for electron transport. The extra electrons are transferred to molecular oxygen, giving rise to superoxide [22,23,24]. Similarly, in our study, the enhanced glucose influx into mesothelial cells cultured under high-glucose conditions resulted in increased ROS production. By reducing glucose influx, both halofuginone and tryptophanol downregulated ROS production. 

In endothelial cells, high-glucose-induced ROS overproduction leads to the inhibition of GAPDH [22,23]. The metabolism of the accumulated, upstream to GAPDH, glycolytic products are diverted into four harmful pathways [27,28]. Similar to endothelial cells, we found that in mesothelial cells, high glucose inhibits GAPDH activity. Both halofuginone and tryptophanol prevented high-glucose-induced GAPDH inhibition. 

In mesothelial cells cultured in a high-glucose medium, we found that the intracellular glucose accumulated due to GAPDH inhibition is diverted to the polyol pathway, as assessed by the cellular D-sorbitol content [23,24]. Both GCN2 kinase activators ameliorated this diversion. The accumulated intracellular fructose 6-phosphate is diverted to the hexosamine pathway, resulting in O-GlcNAc protein modification [23,24]. Interestingly, such a modification of the transcription factor specificity protein-1 (Sp1) upregulates TGF-β1 production [34,35]. We detected an enhanced level of O-GlcNAc-proteins in mesothelial cells cultured under high-glucose conditions, whereas both halofuginone and tryptophanol reduced the level of such proteins. The accumulated fructose 6-phosphate and glyceraldehyde 3-phosphate are diverted to the lipid synthesis pathway. DAG, an intermediate product of this pathway, activates PKC [23,24], which, in turn, by activating nuclear factor kappa-light-chain-enhancer of activated B cells (NF-κB), can upregulate the expression of the profibrotic TGF-β1, as well as various proinflammatory cytokines [36,37,38]. Interestingly, previous studies have shown that high glucose induces PKC activation in peritoneal mesothelial cells, resulting in TGF-β1 and fibronectin production [39], whereas inhibition of PKC-α prevents peritoneal injury in a mouse model of chronic peritoneal exposure to high-glucose dialysate [40]. Our experiments confirmed that, in mesothelial cells cultured under high-glucose conditions, the PKC activity increases, whereas both halofuginone and tryptophanol prevent this to a significant extent. Finally, the accumulated glyceraldehyde 3-phosphate is converted to MGO, a precursor of AGEs [23,24,25]. AGEs exert many effects [41], including PKC activation [42]. In our model, the cellular content of MGO increased in mesothelial cells cultured under high-glucose conditions, while both GCN2 kinase activators inhibited MGO production. Collectively, we confirmed that in mesothelial cells high glucose, by upregulating glucose transporters, enhances glucose influx, oxidative stress, and harmful metabolic pathways, whereas both halofuginone and tryptophanol ameliorate these events.

In PD patients, the degree of submesothelial fibrosis increases over time [9]. In our model, high glucose raised the production of TGF-β1 by mesothelial cells. This is very important since TGF-β1 is the master regulator of extracellular matrix production, as well as for MMT [5,6,7]. The reasons for TGF-β1 upregulation could be many, and the possible role of PKC activation, as well as of O-GlcNAc-Sp1, has been already noted [34,35,37]. Regarding the impact of tryptophanol and halofuginone on TGF-β1 production, we showed that both substances reduce the high-glucose-induced TGF-β1 rise. Halofuginone decreased TGF-β1 production to a greater extent than tryptophanol. This could be attributed to the inhibitory effect of halofuginone on TGF-β1 signal transduction [31], since this cytokine, acting in a paracrine and autocrine fashion, enhances its own production, forming a positive feedback loop [43]. 

Inflammation also contributes to peritoneal membrane injury, and it is well known that in PD patients, frequent episodes of peritonitis accelerate ultrafiltration failure [4]. However, low-grade inflammation is also present even in the absence of infection, possibly due to the toxicity of the commonly used PD solutions [44]. We found that the proinflammatory cytokine IL-8 increases in mesothelial cells cultured in a high-glucose medium. This is in accordance with previous studies [45,46], and may be the result of PKC-induced activation of NF-κB [47]. Also, besides being a mediator and marker of inflammation, studies in cancerous cells revealed that IL-8 is necessary for the epithelial to mesenchymal transition [48]. Whether this cytokine is also necessary for MMT remains to be elucidated. As regards the effect of tryptophanol and halofuginone on IL-8 production, we showed that both substances decrease the high-glucose-induced rise of IL-8. 

Denudation of peritoneal membrane from mesothelial cells characterizes the lesions observed over time in PD treatment [9]. By evaluating the level of activated cleaved caspase-3, in which all the apoptotic pathways converge [49], we showed that high glucose induces mesothelial cell apoptosis. However, both halofuginone and tryptophanol, likely by ameliorating the already described glucotoxic pathways, reduced the high-glucose-induced apoptotic cell death of mesothelial cells, confirming the protective role of GCN-2 kinase activation.

As already noted, in PD, MTT is the primary source of the myofibroblasts [5,6,7], which produce extracellular matrix, causing submesothelial fibrosis. By assessing α-SMA, a marker of differentiation to myofibroblasts [50], we showed that, in mesothelial cells, high glucose induces MTT. Certainly, TGF-β1 plays a significant role in this transition [5,6,7], whereas a role for IL-8 cannot be excluded [48]. Tryptophanol and halofuginone inhibited high-glucose-induced MTT, at least in part due to their effects on TGF-β1 and IL-8 production. MMT downregulation was most noticeable with halofuginone, maybe because halofuginone inhibits TGF-β1 signal transduction as well as activating GCN-2 kinase [31]. 

The in vitro nature of our study is a limitation, but the effect of GCN-2 kinase activation on glucotoxicity in human peritoneal mesothelial cells has never been studied before. We analyzed for the first time the impact of two different GCN-2 kinase activators on the molecular pathways involved in high-glucose-induced glucotoxicity in primary human peritoneal mesothelial cells, and showed a clear protective effect. Thus, our study could be considered a starting point for subsequent in vivo studies aiming to prevent ultrafiltration failure due to the toxicity of the commonly used PD solutions. 

Collectively, our results are depicted in Figure 8 and indicate that the activation of GCN-2 kinase, at least in part by inhibiting glucose entry into the cell, ameliorates glucotoxicity in primary human peritoneal mesothelial cells, preserves their integrity, and prevents MMT. Whether such a strategy could be applied in the clinic to avoid ultrafiltration failure in PD patients remains to be investigated.

## Figures and Tables

**Figure 1 biomolecules-09-00832-f001:**
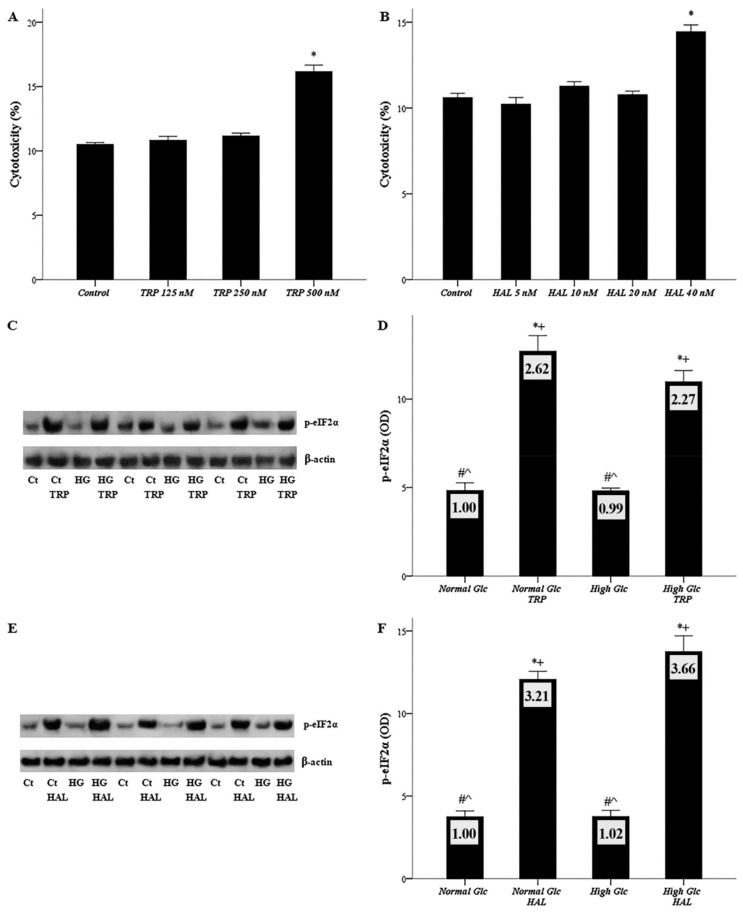
Both halofuginone and tryptophanol, at nontoxic concentrations, activate GCN-2 kinase. In mesothelial cells cultured under normal glucose, tryptophan at a concentration of 250 nM, and halofuginone at a concentration of 20 nM were not cytotoxic. Thereafter, these concentrations were used for all subsequent experiments (**A**,**B**). The ability of the above substances at the above concentrations to activate GCN-2 kinase was evaluated by the level of phosphorylation of the GCN-2 kinase substrate e-IF2α with Western blotting. Nine experiments were performed for each substance, and three of them are depicted in (**C**) and (**E**). In mesothelial cells cultured under normal or high-glucose conditions, both halofuginone and tryptophanol activated GCN-2 kinase (**D**,**F**). *, #, +, and ^ indicate *p* < 0.05 compared to the first, second, third, or fourth depicted conditions. Error bars correspond to standard error of means. In the plots of the WB results, the number inside each bar corresponds to the mean fold-change compared to the control.

**Figure 2 biomolecules-09-00832-f002:**
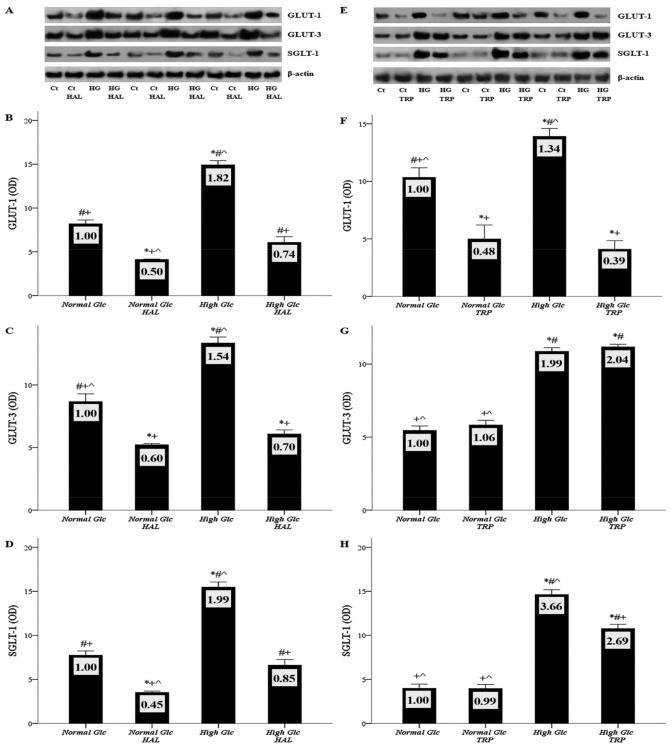
The effect of high glucose and halofuginone or tryptophanol on glucose transporters. Mesothelial cells were cultured under normal or high-glucose conditions in the presence or not of halofuginone or tryptophanol. GLUT-1, GLUT-3, and SGLT-1 levels were assessed with Western blotting. Nine experiments were performed for each substance; three of them are depicted in panels (**A**) and (**E**). High glucose increased the expression of all glucose transporters (**A**–**H**). Halofuginone reduced the expression of GLUT-1, GLUT-3, and SGLT-1 (**B**–**D**). Tryptophanol decreased the level of GLUT-1 (**F**), and, to a lesser extent, SGLT-1 (**H**), while it left GLUT-3 unaffected (**G**). *, #, +, and ^ indicate *p* < 0.05 compared to first, second, third, or fourth depicted conditions. Error bars correspond to standard error of means. The number inside each bar corresponds to the mean fold-change compared to the control.

**Figure 3 biomolecules-09-00832-f003:**
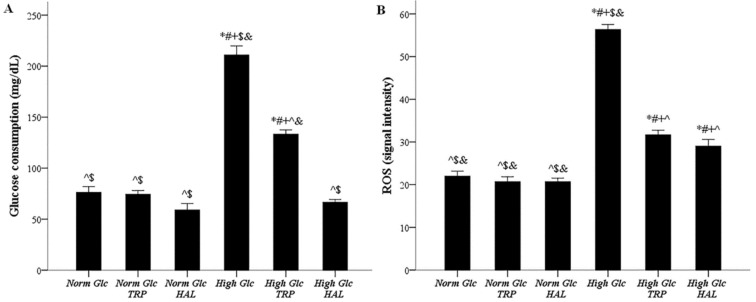
The impact of high glucose and tryptophanol or halofuginone on glucose consumption and ROS production. Mesothelial cells were cultured under normal or high-glucose conditions with or without halofuginone or tryptophanol. Under high-glucose conditions, mesothelial cells consumed more glucose and produced more ROS (**A**,**B**). Both tryptophanol and halofuginone reduced glucose consumption, halofuginone to a greater extent (**A**), and ROS production (**A**,**B**). *, #, +, ^, $, and & indicate *p* < 0.05 compared to the first, second, third, fourth, fifth, or sixth depicted conditions. Error bars correspond to standard error of means.

**Figure 4 biomolecules-09-00832-f004:**
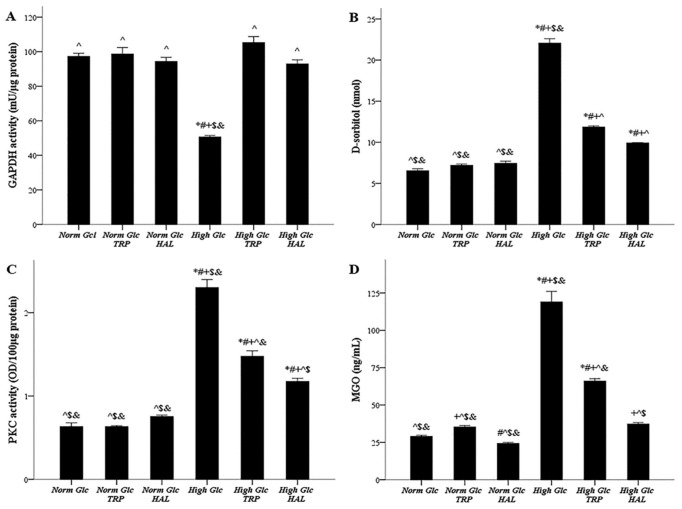
The effect of high glucose and tryptophanol or halofuginone on GAPDH activity, aldose pathway, PKC activity, and AGEs generation pathway. Mesothelial cells were cultured under normal or high-glucose conditions in the presence or not of halofuginone or tryptophanol. High glucose inhibited GAPDH activity, while both tryptophanol and halofuginone restored its activity (**A**). High glucose diverted metabolism towards the aldose pathway, as assessed by the cellular D-sorbitol. Both tryptophanol and halofuginone ameliorated this high-glucose-induced alteration (**B**). High glucose activated PKC, while both tryptophanol and halofuginone antagonized this effect (**C**). High glucose promoted AGEs production, assessed by cellular MGO, whereas tryptophanol and halofuginone decreased MGO, halofuginone to a greater extent than tryptophanol (**D**). *, #, +, ^, $, and & indicate *p* < 0.05 compared to first, second, third, fourth, fifth, or sixth depicted conditions. Error bars correspond to standard error of means.

**Figure 5 biomolecules-09-00832-f005:**
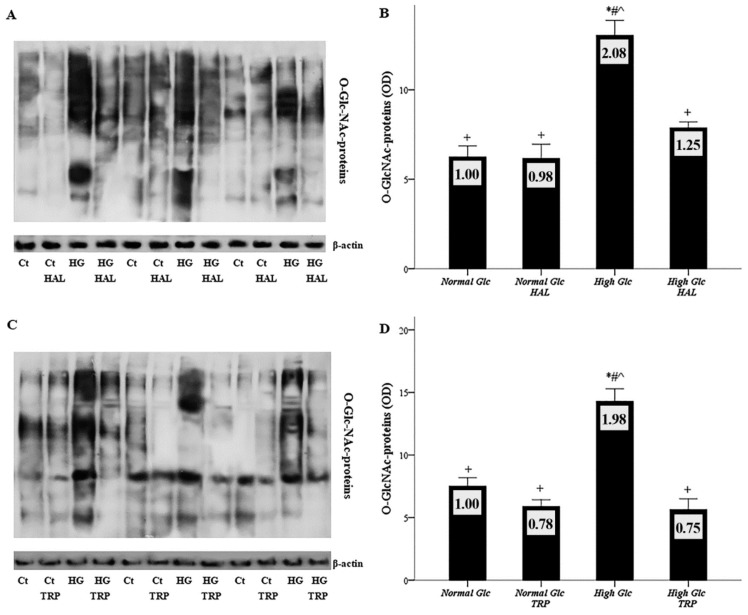
The effect of high glucose and tryptophanol or halofuginone on hexosamine pathway. Mesothelial cells were cultured under normal or high-glucose conditions, with or without halofuginone or tryptophanol. Hexosamine pathway was assessed by the level of O-GlcNAc-proteins with Western blotting. Nine experiments were performed for each GCN-2 kinase activator; three of them are depicted in panels (**A**) and (**C**). High glucose upregulated the level of O-GlcNAc-proteins (**B**,**D**), whereas both halofuginone (**B**) and tryptophanol (**D**) restored the O-GlcNAc-proteins level to that observed under normal glucose. *, #, +, and ^ indicate *p* < 0.05 compared to first, second, third, or fourth depicted conditions. Error bars correspond to standard error of means. The number inside each bar corresponds to the mean fold-change compared to the control.

**Figure 6 biomolecules-09-00832-f006:**
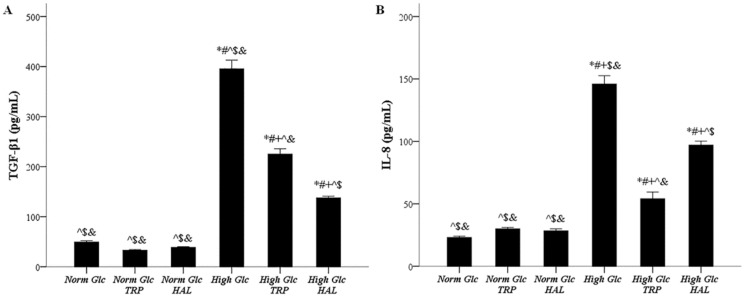
The impact of high glucose and tryptophanol or halofuginone on TGF-β1 and IL-8 production. Mesothelial cells were cultured under normal or high-glucose conditions in the presence or not of halofuginone or tryptophanol. TGF-β1 and IL-8 production was assessed by their concentrations in cell culture supernatants. High glucose enhanced TGF-β1 level, tryptophanol reduced the degree of high-glucose-induced TGF-β1 increment, and halofuginone exerted the same effect, to a greater extent (**A**). High glucose increased IL-8 concentration; tryptophanol reduced the degree of high-glucose-induced IL-8 rise, followed by halofuginone to a lesser degree (**B**). *, #, +, ^, $, and & indicate a *p* < 0.05 compared to first, second, third, fourth, fifth, or sixth depicted conditions. Error bars correspond to standard error of means.

**Figure 7 biomolecules-09-00832-f007:**
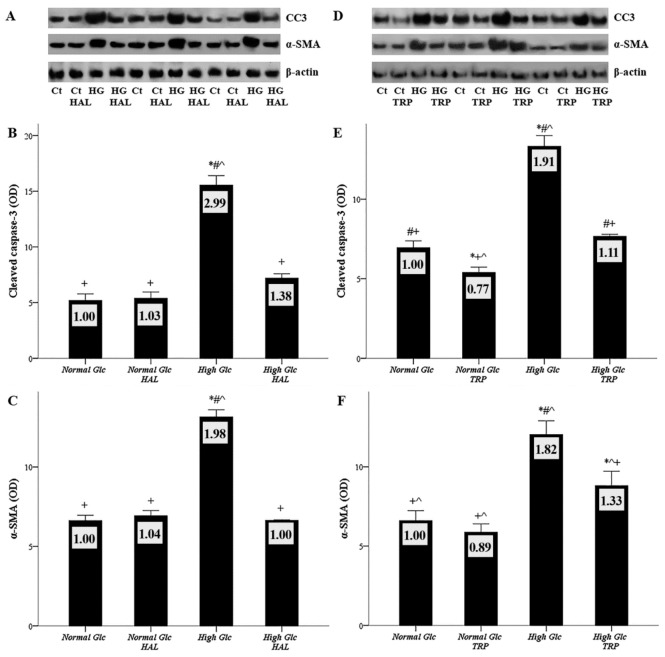
The effect of high glucose and tryptophanol or halofuginone on mesothelial cell apoptosis and mesothelial to mesenchymal transition. Mesothelial cells were cultured under normal or high-glucose conditions, with or without halofuginone or tryptophanol. Apoptosis was assessed by the level of activated cleaved caspase-3, and MMT by the level of α-SMA with Western blotting. Nine experiments were performed for each GCN-2 kinase activator; three of them are depicted in panels (**A**) and (**D**). High-glucose conditions induced apoptosis (**B**,**E**), while both halofuginone (**B**) and tryptophanol (**E**) prevented apoptosis. High-glucose conditions caused MMT (**C**,**F**), while both halofuginone (**C**) and tryptophanol (**F**) prevented MMT, the latter to a lesser extent than halofuginone. *, #, +, and ^ indicate *p* < 0.05 compared to first, second, third, or fourth depicted conditions. Error bars correspond to standard error of means. The number inside each bar corresponds to the mean fold-change compared to the control.

**Figure 8 biomolecules-09-00832-f008:**
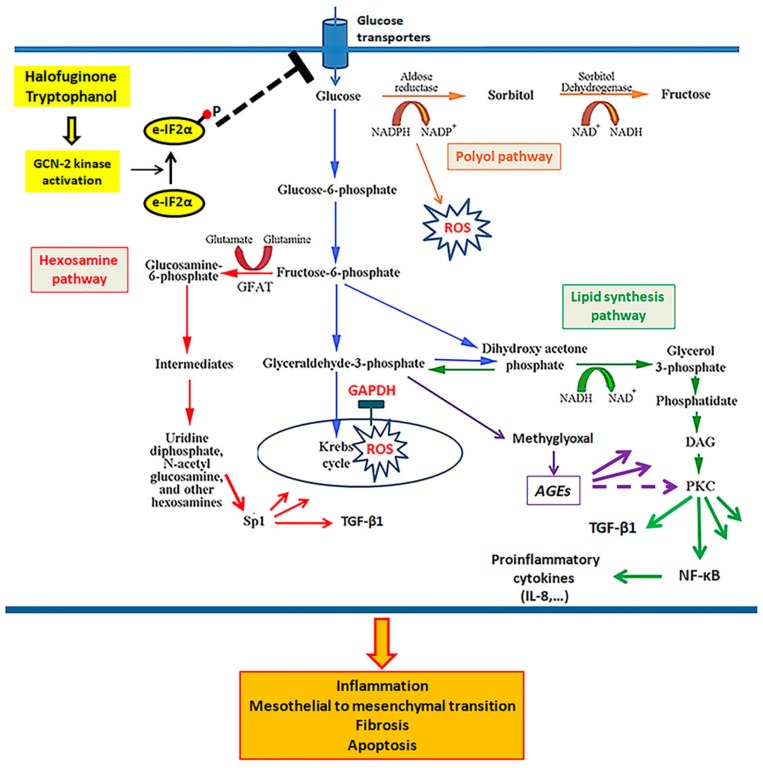
The effect of GCN-2 kinase activation on glucotoxicity, cell integrity, and the mesothelial to mesenchymal transition in human peritoneal mesothelial cells. Under high-glucose conditions, glucose influx into mesothelial cells increases, likely due to the upregulation of glucose transporters. The elevated intracellular glucose is catabolized through the Krebs cycle and produces more electron donors (NADH and FADH_2_) than the oxidative phosphorylation chain has capacity to transfer electrons. The extra electrons convert molecular oxygen to superoxide. ROS inhibits the enzyme GAPDH, leading to accumulation of upstream glycolytic products, which are diverted to four noxious pathways. The polyol pathway utilizes NADPH, required for the reduction of glutathione. The hexosamine pathway results in the O-GlcNAc-modification of proteins, altering their function. For instance, the modification of Sp1 increases the transcription of the gene for the profibrotic TGF-β1. The lipid synthesis pathway produces DAG, an activator of PKC kinase able to activate NF-κB, known for its role in the expression of pro-inflammatory cytokines and TGF-β1. Finally, glyceraldehyde 3-phosphate is converted to MGO, a precursor of AGEs. All these glucotoxic events increase the production of the profibrotic TGF-β1 and pro-inflammatory cytokines, such as IL-8, and induce cell apoptosis and the mesothelial to mesenchymal transition—alterations involved in the peritoneal membrane pathology observed in PD patients with ultrafiltration failure. Activation of GCN-2 kinase by downregulating the expression of glucose transporters and glucose influx into the mesothelial cells prevents all of the above described glucotoxic pathways, preserves cell integrity, and prevents mesothelial to mesenchymal transition.

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
