# Peer review of "Activation of General Control Nonderepressible-2 Kinase Ameliorates Glucotoxicity in Human Peritoneal Mesothelial Cells, Preserves Their Integrity, and Prevents Mesothelial to Mesenchymal Transition"

_biomolecules, 2019, doi:10.3390/biom9120832_

Round 1

Reviewer 1 Report

Comment on the manuscript biomolecules-624481 
This manuscript describes that halofuginone of 20 nM or tryptophanol of 250 nM will activate GCN-2 kinase under normal or high glucose condition in primary human peritoneal mesothelial cells. This research is important in preclinical research using halofuginone or tryptophanol to inhibit all the high glucose-induced alterations, including activation of GCN-2 kinase to ameliorate glucotoxicity, preservation of their integrity, and prevention of MMT. Although the paper is interesting, it also includes insufficient points as described below.

It is interesting to know whether the halofuginone + tryptophanol could induce addition reaction to affect the GCN-2 kinase activation and functionality in peritoneal mesothelial cells.

Tryptophanol, also called indole-3-ethanol, is a dietary indole present in cruciferous vegetables that has been shown to influence estradiol metabolism in humans and may provide a new chemo-preventive approach to estrogen-dependent diseases. (PMID 2342128). Since tryptophanol didn’t affect GLUT-3 protein expression (Figure 2), would it be possible that tryptophenal may work through another pathway in addition to GCN-2 kinase activation?

What is the difference in the concentration of low glucose or normal glucose in the material section of this manuscript?

Please make the straight coordinates at a similar scale to clarify the meaning, for example: Fig1, Fig5.

Please use more updated references, such as the papers on the relationship between high glucose-induced toxicity and peritoneal mesothelial cells or renal disease.

Please confirm whether the effect of AGEs on PKC activation is solid or dotted line in Fig.8.

Author Response

Firstly, we would like to thank the reviewer since his/her valuable comments helped us to clarify specific points in our manuscript and improve on its quality.

It is interesting to know whether the halofuginone + tryptophanol could induce addition reaction to affect the GCN-2 kinase activation and functionality in peritoneal mesothelial cells.

Our aim was not to compare halofuginone, which is already in use for different reasons in veterinary medicine and clinical practice, with tryptophanol, an experimental compound. Both substances activate GCN2 kinase, and this was the scope of their use. Thus, the use of the two substances in our study was done for confirmatory reasons, and also, because as already noted in the text, halofuginone besides GCN2 kinase activation interferes with TGF-β signal transduction.

Tryptophanol, also called indole-3-ethanol, is a dietary indole present in cruciferous vegetables that has been shown to influence estradiol metabolism in humans and may provide a new chemo-preventive approach to estrogen-dependent diseases. (PMID 2342128). Since tryptophanol didn’t affect GLUT-3 protein expression (Figure 2), would it be possible that tryptophenal may work through another pathway in addition to GCN-2 kinase activation?

This is a common misunderstanding. Indole-3-ethanol by mistake in some sites, such as the http://www.hmdb.ca/metabolites/HMDB0003447 is referred to as tryptophanol, which is wrong. Indole-3-ethanol is not tryptophanol but tryptophol. The chemical structure of two compounds differs. The possible reason for some differences in the effects of halofuginone and tryptophanol of mesothelial cells has been already noted in the text, and it is the fact that halofuginone also interferes with TGF-β signal transduction. 

What is the difference in the concentration of low glucose or normal glucose in the material section of this manuscript?

As noted in the second paragraph low glucose culture medium contains normal concentration of D-glucose (5.5 mM), whereas high glucose culture medium contains 25 mM of D-glucose. This is clarified better in the revised manuscript. The L-glucose was added in low glucose culture medium to achieve equal osmolarity. L-glucose is not metabolized by the cells. The latter is also noted in the revised manuscript.

Please make the straight coordinates at a similar scale to clarify the meaning, for example: Fig1, Fig5.

In the revised manuscript, the plots that correspond to WB changed, and a number was added in each bar that depicts the mean fold-change compared to the control to make comparisons easier.

Please use more updated references, such as the papers on the relationship between high glucose-induced toxicity and peritoneal mesothelial cells or renal disease.

More recent references about high glucose-induced toxicity in peritoneal mesothelial cells are added in the revised manuscript.

Please confirm whether the effect of AGEs on PKC activation is solid or dotted line in Fig.8. 

The line is dotted to indicate that AGEs may activate PKC after many steps. There are data supporting that AGEs activate PKC through activation of RAGEs. We did not show the entirety of the pathway in the figure since it would make the figure extremely cumbersome.

Reviewer 2 Report

Very well designed and performed study.

I want to congratulate the authors for this very nice and important piece of work in the peritoneal dialysis field.

The use of the word MAY could be applied in some of the assumptions/conclusions made. For example, in the abstract "Many studies confirmed that high glucose concentration, GDPs, and AGEs may induce MMT [5-7]." It is not clear if the primary human peritoneal mesothelial cells came from healthy subjects (most likely) or from uremic patients before or already undergoing dialysis therapy. It is known that not only peritonitis and high glucose load determine the peritoneal membrane structural changes that may lead to functional changes (ultrafiltration loss, reduction in solute clearance). Uremia per se and longevity of the dialysis therapy may also play a role. John Williams et al have brought up this discussion in their papers ( 2002 JASN paper and 2003 Kidney International). Their data (Biopsy Registry) suggested that morphological changes of the peritoneal membrane begin before the start of renal replacement therapy; and these changes appear to progress during hemodialysis. I do agree that the in vitro nature of their study is a limitation, but this is an original and novel study that needs to be shared with the peritoneal dialysis community and may lead to further development in the PD therapy, with benefits to everyone involved with the PD therapy  In the discussion section, I do not see any comments about other potential strategies to reduce glucotoxicity in mesothelial cells, such as different osmotic agents ( icodextrin, xylitol, glycerol, sorbitol).

Author Response

Firstly, we would like to thank the reviewer for his/her encouraging comments. Also, these clinically oriented comments helped us to make our manuscript more accurate and improved.

The use of the word MAY could be applied in some of the assumptions/conclusions made. For example, in the abstract "Many studies confirmed that high glucose concentration, GDPs, and AGEs may induce MMT [5-7]."

As recommended, the word “may” was added in some assumptions/conclusions of our manuscript.

It is not clear if the primary human peritoneal mesothelial cells came from healthy subjects (most likely) or from uremic patients before or already undergoing dialysis therapy. It is known that not only peritonitis and high glucose load determine the peritoneal membrane structural changes that may lead to functional changes (ultrafiltration loss, reduction in solute clearance). Uremia per se and longevity of the dialysis therapy may also play a role. John Williams et al have brought up this discussion in their papers ( 2002 JASN paper and 2003 Kidney International). Their data (Biopsy Registry) suggested that morphological changes of the peritoneal membrane begin before the start of renal replacement therapy; and these changes appear to progress during hemodialysis.

Thank you for this valuable comment. In the revised manuscript, we added the related information.

In the discussion section, I do not see any comments about other potential strategies to reduce glucotoxicity in mesothelial cells, such as different osmotic agents ( icodextrin, xylitol, glycerol, sorbitol). 

In the revised manuscript, a comment about the attempts to reduce glucotoxicity by using glucose-free peritoneal dialysis solutions is added.

Round 2

Reviewer 1 Report

The authors have adequate responses to the reviewer's comment. The manuscript is acceptable for publication.